# Reviewing the Regulators of COL1A1

**DOI:** 10.3390/ijms241210004

**Published:** 2023-06-11

**Authors:** Hanne Devos, Jerome Zoidakis, Maria G. Roubelakis, Agnieszka Latosinska, Antonia Vlahou

**Affiliations:** 1Centre of Systems Biology, Biomedical Research Foundation of the Academy of Athens, 11527 Athens, Greece; hdevos@bioacademy.gr (H.D.); izoidakis@bioacademy.gr (J.Z.); 2Laboratory of Biology, University of Athens School of Medicine, 11527 Athens, Greece; roubel@med.uoa.gr; 3Laboratory of Cell and Gene Therapy, Biomedical Research Foundation, Academy of Athens, 11527 Athens, Greece; 4Mosaiques Diagnostics GmbH, 30659 Hannover, Germany

**Keywords:** collagen, collagen type I alpha 1 chain, *COL1A1*, molecular signalling pathways, disease

## Abstract

The collagen family contains 28 proteins, predominantly expressed in the extracellular matrix (ECM) and characterized by a triple-helix structure. Collagens undergo several maturation steps, including post-translational modifications (PTMs) and cross-linking. These proteins are associated with multiple diseases, the most pronounced of which are fibrosis and bone diseases. This review focuses on the most abundant ECM protein highly implicated in disease, type I collagen (collagen I), in particular on its predominant chain collagen type I alpha 1 (COLα1 (I)). An overview of the regulators of COLα1 (I) and COLα1 (I) interactors is presented. Manuscripts were retrieved searching PubMed, using specific keywords related to COLα1 (I). *COL1A1* regulators at the epigenetic, transcriptional, post-transcriptional and post-translational levels include DNA Methyl Transferases (DNMTs), Tumour Growth Factor β (TGFβ), Terminal Nucleotidyltransferase 5A (TENT5A) and Bone Morphogenic Protein 1 (BMP1), respectively. COLα1 (I) interacts with a variety of cell receptors including integrinβ, Endo180 and Discoidin Domain Receptors (DDRs). Collectively, even though multiple factors have been identified in association to COLα1 (I) function, the implicated pathways frequently remain unclear, underscoring the need for a more spherical analysis considering all molecular levels simultaneously.

## 1. Introduction

The most abundant proteins in the mammalian proteome are collagens, constituting roughly 30% of the total protein mass [1], typically being present in the extracellular matrix (ECM; a list of all abbreviations in this review is shown in Appendix A, Table A1) where they contribute to tissue structure [2]. Currently, 28 proteins are part of the collagen protein superfamily, denoted by Roman numerals, and characterized by at least one triple-helix domain [1]. Most collagens, such as type I collagen (collagen I), type II collagen (collagen II) and type III collagen (collagen III) (fibrillar collagens) [1,2], are ubiquitously expressed across tissues, whereas others [for example type VII collagen (collagen VII)], are more tissue-specific and have a function related to the tissue of origin (for example collagen VII’s role in dermal–epidermal adhesion) [1].

On the molecular level, all collagens consist of three left-handed polyproline helices (chains), combined in one right-handed triple helix (Figure 1). Each chain consists of repeats of Gly-X-Y, with Gly denoting a glycine residue, X frequently denoting a proline (Pro) residue, and Y frequently denoting a 4-hydroxyproline residue [1]. For pro-collagen I, the triple helix is surrounded by C-terminal and N-terminal propeptides [3] (Figure 1), which lack the typical Gly-X-Y structure and are cleaved during maturation to collagen I (see below) [4,5]. The type of chains interacting to form a collagen molecule depends on the collagen type, with the most abundant collagen I consisting of two collagen type I alpha 1 (COLα1 (I)) and one collagen type I alpha 2 (COL1α2 (I)) chains (Figure 1) [1,3].

Collagen aberrations have been associated with multiple diseases, with the most prominent being fibrosis, characterized by disbalance between collagen synthesis and degradation leading to excess collagen deposition [17]. In lung fibrosis more specifically, increased levels of COLα1 (I), produced by alveolar macrophages, have been associated with disease progression [18]. In addition to the involvement of collagen in fibrosis, COLα1 (I) is also involved in genetic diseases. For example, lethal mutations in osteogenesis imperfecta (OI) were linked to binding sites for integrins and glycosaminoglycans on collagen I [1]. 

Among the different collagen members, collagen I is the most abundant, with its levels corresponding to more than 90% of the ECM [19]. This review focuses on COLα1 (I), the predominant chain in collagen I [1]. Following a brief introduction on its structure, we specifically investigate in depth the recent literature on COLα1 (I), placing special emphasis on the mechanisms of its regulation as well as its cell surface interactors and induced pathways in health and disease including but not limited to fibrosis.

### COL1A1 Synthesis and Basic Structure

Following its synthesis human pro-COLα1 (I) has a total length of 1464 amino acids, of which amino acids 1–22 denote a signal peptide, amino acids 23–161 an N-terminal propeptide, amino acids 162–1218 the mature collagen chain and amino acids 1219–1464 a C-terminal propeptide [20]. Within the N-terminal propeptide, a von Willebrand factor type C domain (vWFC) exists (amino acids 38–96), while within the C-terminal propeptide, the Fibrillar collagen C-terminal non-collagenous domain (NC1, amino acids 1229–1464) is located [20], the latter also being denoted as COLF [21] (Figure 1). Various post-translational modifications (PTMs) then occur intracellularly [1] which are involved in the formation of the mature COLα1 (I) chain: these include the formation of disulfide bonds within the pro-peptide extensions, isomerization of peptidyl-prolyl bonds, hydroxylation of lysyl (Lys) and Pro residues, and glycosylation of several hydroxylysines (Figure 1). The most common form of prolyl hydroxylation is prolyl-4-hydroxylation, occurring on roughly half of the prolyl residues. Prolyl-3-hydroxylation is also observed, in one residue per chain, Pro986 on the (pro-)COLα1 (I) and Pro707 on the (pro-)COLα2 (I) [14,16]. Despite its rare occurrence, the latter modification is well-conserved across species. Hydroxylation is considered important for the establishment of a strong fibril of interacting collagen molecules on the supramolecular level [14]. While hydroxyproline is important for the formation of triple-helical domains, hydroxylysyl regulates the formation of cross-links and glycosylation [15].

Following the production of the individual chains, the formation of the pro-collagen I triple helix then follows. As mentioned before, (pro-)collagen I is primarily a heterotrimer consisting of two (pro-)COLα1 (I) chains and one (pro-)COLα2 (I) chain [3]. The COLF C-terminal domain of COLα1 (I) is critical to maintain the correct chain combination (e.g., two COLα1 (I) chains and one COLα2 (I) chain) and orientation before fibril formation in the ECM [5]. The N-terminal region of COLα1 (I) has been associated with the formation of heterogenous fibrils, transcellular transport and secretion, proteolytic processing, feedback regulation of synthesis and fibrillogenesis [22]. Interestingly, COLα1 (I) homotrimers have also been detected. Such homotrimers are associated with structural and functional differences to the classical heterotrimer forms, for example an increased resistance against proteases [23], weaker intermolecular interactions leading to reduced tensile strength [24], and an increased lateral space [25].

After the combination of the three separate pro-collagen I chains and post-translational modifications intracellularly, stabilization of collagen I and later, collagen I fibrils (consisting of multiple cross-linked collagen I molecules), mediated by cross-linking, occurs in the ECM [3,14]. Three different ways of cross-linking have been described [1]: (A) cross-linking via N^ε^(γ-glutamyl)lysine, which occurs between the amino groups of glutamine and of lysine [26] by transglutaminase (Figure 2A); (B) cross-linking via the lysyl oxidase (LOX) pathway. After oxidative deamination to peptidyl-α-aminoadipic-δ-semialdehyde mediated by LOX, two hydroxylysine residues are cross-linked through a spontaneous condensation reaction, as shown in Figure 2B. This type of cross-linking occurs both between α-chains within one collagen I molecule and between α-chains of different collagen I molecules and (C) occasionally cross-linking may also be mediated by Advanced Glycation End products (AGE), as shown in Figure 2C. In the presence of increased levels of AGE, the consequent increased collagen I cross-linking results in more brittle fibrils, thus providing a link between collagen fibril stability and increased sugar levels [27]. Cross-linking is overall considered a vital step for the formation of collagen fibrils on the supramolecular level, protecting from degradation [28]. Similar to collagen protein expression, cross-linking is also known to vary across tissues. These differences seem less linked to the specific tissue, and more to tissue function [15]. Lysine aldehyde cross-linking occurs more frequently in soft tissues such as the skin, while hydroxylysine occurs more frequently in connective tissues related to the skeletal system. Such differences were hypothesized to be due to differential expression of the lysyl hydroxylation genes in these tissues [29]. Moreover, collagen I fibrils in the periodontal ligament showed reduced lysine 3-hydroxylation, and preference for hydroxylysine aldehyde cross-linking, in comparison to collagen molecules found in other ligaments or skin, where lysine aldehyde-based cross-linking is preferred [30].

Collagen maturation includes cleavage by proteases: proteases cleaving pro-COLα1 (I) giving rise to the mature collagen I molecule are A Disintegrin And Metalloproteinase with Thrombospondin motifS 2 (ADAMTS2), cleaving at amino acids 161–162, resulting in cleavage of the N-terminal propeptide; and Bone Morphogenic Protein (BMP1), cleaving at amino acids 1218–1219, resulting in cleavage of the C-terminal propeptide (Figure 3). BMP1 is in turn regulated by pro-collagen C-proteinase enhancer 1 (PCPE-1) [3].

Collagen I as well as other collagens in the ECM is degraded by matrix-metalloproteinases (MMPs), zinc-dependent multi-domain endopeptidases containing a signal peptide targeting them for secretion, a pro-peptide domain which is cleaved off leading to MMP activation, and a highly conserved Zn^2+^-binding motif. The family of MMPs counts 24 members in humans with collagen I (amongst others) being cleaved by MMP-1, MMP-8, MMP-13 and MMP-25 (Figure 3) [36,37]. In addition, MMP-1 and MMP-2 cleave incorrectly synthesized collagens, or partially degraded collagens [36,38]. Moreover, MMP-9 is known to cleave COLα1 (I) resulting in two fragments of ¾ and ¼ of COLα1 (I)’s length, although the exact cleavage site remains to be confirmed [38]. Typically, expression of MMPs has to be induced, as there is no constitutive expression of these proteins. Regulation occurs on the level of transcription [via transcription factors such as Activator Protein 1 (AP1)], post-translational modifications (proteolytic cleavage) and cellular localization (in the case of MMP-2 and MMP-8 binding to cell receptors is required to keep them localized) [36]. Moreover, a specific cellular localization separates the MMPs from the Tissue Inhibitors of Matrix metalloproteinases (TIMPs), which inhibit the enzymatic function of MMPs [36]. Besides MMPs, Cathepsin S also cleaves collagen I in three different positions (Figure 3) [37]. Collectively, (pro-)COLα1 (I) synthesis, PTMs, incorporation into mature collagen I fibrils and further superstructures and ultimately degradation, are tightly regulated context-dependent processes.

**Figure 3 ijms-24-10004-f003:**
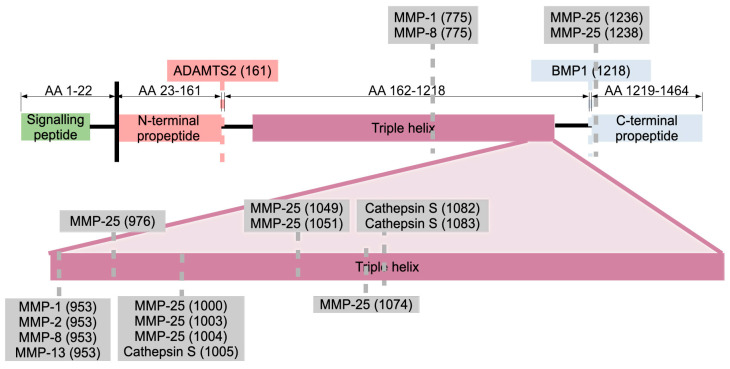
Proteases cleaving the COLα1 (I) amino acid chain. For each protease and cleavage site, the information is shown as follows: protease name (number of the C-terminal amino acid after cleavage). ADAMTS2 cleaves off the N-terminal propeptide (pink) [2] and BMP1 cleaves off the C-terminal propeptide (blue) [2]. Other cleavage sites are indicated in grey, mainly MMP-25 and three sites for Cathepsin S [37]. Only proteases with characterized cleavage sites based on the literature [38], the UniProtKB website [20] and the Proteasix website [37] (supported by literature evidence) are shown. Abbreviations are explained in Appendix A, Table A1.

## 2. Literature Search and Results

### 2.1. Literature Search

A literature search was carried out, using the keywords (Medical Subject Headings or MeSH terms) “col1a1”, “collagen type I alpha 1” or “col1a1 protein human”, in combination with one or more of the following MeSH terms: “signalling pathways”, “disease”, “protein function”, “protein structure”, “chronic disease”, “pathological process”, “chronic disease”, “post-translational protein processing”, “function”, as summarized in Figure 4. For all searches no time limit was used (with the only exceptions of “col1a1” and “disease” search with applied time limit of 2017–2022). Collectively, this search led to the identification of 984 potentially relevant articles (Figure 4).

These were screened for relevance to COLα1 (I) based on their abstract, excluding reviews. This analysis generated a shorter list of 918 manuscripts which were further evaluated based mainly on their abstract, introduction and conclusion. Following application of the additional exclusion criteria (listed on Figure 4; details provided in Appendix A, Table A2) a final list of 64 articles was retained, forming the backbone of this review (Appendix A, Table A3), and summarized below.

### 2.2. Regulators of COL1A1 and COLα1 (I)

The synthesis and maturation of COLα1 (I) is regulated on multiple levels: the epigenetic and transcriptional, the post-transcriptional, as well as the post-translational levels. A summary of the existing evidence, as compiled based on our literature search, is provided below. Figure 5 below provides an overview of our findings, while additional information (such as model used, disease implicated and citation) can be found in Appendix A, Table A4.

#### 2.2.1. COL1A1 Gene Epigenetic and Transcriptional Regulators

Two epigenetic regulators (DNA Methyl Transferases 1 and 3A or DNMT1 and DNMT3A) were identified to regulate genes in turn regulating *COL1A1* (indirect regulation). Both belong to the DNMT family and were investigated in the context of liver fibrosis [39,40,41]. Additionally, HDAC1 and KDM1A have also been identified to directly regulate *COL1A1* in the context of the TGFβ pathway [42,43].

Increased *COL1A1* mRNA expression in bronchoalveolar lavage cells from human patients suffering from idiopathic pulmonary fibrosis and lung cancer is presumed to be the consequence of DNMT1 regulation, although the underlying mechanism remains to be elucidated [41]. In the case of liver fibrosis, which was investigated in a respective mouse model (fibrosis induced using carbon tetrachloride (CCl_4_)), DNMT1 was hypothesised to reduce *Pten* mRNA levels through hypermethylation of its promotor, indirectly influencing COLα1 (I) expression levels [39].

Along these lines, an increase in *Col1a1* mRNA expression was also observed in a rat model of liver fibrosis established by administering CCl_4_ in comparison to healthy controls. In the same model, DNA Methyl Transferase 3A (DNMT3A) was upregulated and its inhibition could prevent the aforementioned increase in *Col1a1* mRNA. Moreover, overexpression of Antisense non-coding RNA in the INK4 locus (ANRIL) prevented the increase in *Col1a1* mRNA, while conversely, repressing ANRIL had the opposite effect. Similar changes have been observed in human liver samples [40]. This implies that DNMT3A and ANRIL regulate *Col1a1* mRNA levels, although their exact mechanism of action was not described [40].

More evidence is available on the regulation of *COL1A1* at the transcriptional level. Our literature search highlighted transcription factors with existing evidence for direct binding to the upstream regions of *COL1A1*, namely Nuclear factor kappa-light-chain-enhancer of activated B cells (NF-κB) [44] and Sma and Mad related protein 2/3 complex (Smad2/3 complex) in the context of the TGFβ signalling pathway [45]. In addition, direct evidence exists for Nuclear Receptor Subfamily 4 Group A Member 1 (NR4A1) [42], Specific Protein 1 (SP1), Activating enhancer binding Protein 2 (AP2) [46,47] and Krüppel-like factor 6 (KLF6) [48]. Moreover, indirect evidence linking c-Myb [49], Neurogenic locus notch homolog protein 2 (NOTCH2) [50], Hypoxia Inducible Factor 1 Subunit Alpha (HIF1α) [51], the Yes-associated proteins (YAP)/Tafazzin (TAZ) proteins [52] and Zinc finger E-box-binding homeobox 1 (ZEB1) [53] to *COL1A1* transcription have been found. A brief overview of the existent evidence is provided below.

The transcription factor NF-κB has been studied the most extensively in association to *Col1a1* regulation, specifically in murine endothelial fibroblasts [44]. This transcription factor is a heterodimer, consisting of Rel-associated protein (RelA) or p65, and p50. It has four binding regulatory loci on the *Col1a1* gene: at −8618 base pairs (bp), −7831 bp, −7082 bp and +4 bp. NF-κB is recruited to binding site −8618 bp by the activation of the Receptor for Advanced Glycation End products (RAGE) by AGE, reflecting a link of *Col1a1* mRNA synthesis with increased glucose levels [44]. NF-κB’s effect is related to its PTMs. Of relevance here are the phosphorylation of Serine (Ser) 311 and Ser 536 (for *Col1a1* and *Col1a2*) and threonine (Thr) 254 (for *Col1a2*). Mutations of the residues Ser 536 and Thr 254 affect binding to the 5′ regions of the *Col1a1* and *Col1a2* genes, respectively, in murine endothelial fibroblasts [44].

One frequently cited pathway in *COL1A1* regulation is the TGFβ pathway [45]; TGFβ activates the TGFβ receptor, with pleiotropic downstream effectors (reviewed in Trelford et al. [54]). Of the most known is Smad2/3 [55], which upon its activation plays a central role in the transcriptional regulation of multiple fibrosis-related genes, including *COL1A1*. In brief, the phosphorylated Smads bind Smad4, translocate to the nucleus, and induce gene expression of, amongst others, collagens and collagen receptors such as integrins [45] (see also Section 2.3). A transcriptional complex of Smad3 and β-catenin has been additionally suggested in the context of *COL1A1* transcriptional regulation in chronic obstructive pulmonary disease as silencing of β catenin prevented a TGFβ-mediated increase in *COL1A1* mRNA and COLα1 (I) levels in a human airway epithelial cell line [56]. However, the existence of such a complex has only been experimentally confirmed in the context of αSMA regulation in pulmonary alveolar cells [57]. Besides Smads, AP2 has also been found to bind the *COL1A1* promotor in TGFβ-challenged dermal fibroblasts from systemic sclerosis patients [46,47]. The TGFβ-induced downstream pathways vary with a consensus on the involvement of p42/p44 Mitogen-Activated Protein Kinase (MAPK). Increased phosphorylation of p38 was associated with increased *COL1A1* mRNA levels, in in vitro systems related to pulmonary veno-occlusive disease [58] and liver fibrosis [59]. The downstream interactors of p38 affecting *COL1A1* mRNA expression were, however, not described [58,59]. Another TGFβ downstream pathway that has been suggested as implicated in *COL1A1* regulation is the phosphoinositide 3 kinase (PI3K)/Akt /mTOR (Mammalian Target of Rapamycin) pathway; this was suggested by studies on human lung fibroblast cells from pulmonary fibrosis patients treated with isoliquiritigenin (a natural flavonoid, with anti-inflammatory and anti-oxidative properties), resulting in a decrease in COLα1 (I), which was more pronounced in the presence of a PI3K/Akt inhibitor [60].

Further signalling pathways influencing *COL1A1* transcription via (at least to some extent) TGFβ are Mer Tyrosine Kinase (MERTK) [61], PKCε [62], Ovarian cancer G-protein coupled Receptor-1 (*OGR1*) [63], β-catenin [56], and osteopontin (more details can be found in Appendix A, Table A5). In brief, reduced mRNA levels of amongst others *Col1a1* have been reported in the liver of a mouse model of non-alcoholic fatty liver disease (NAFLD) upon injection with M2c macrophages expressing MERTK [61], which has been suggested to induce TGFβ secretion in the context of Immunoglobulin G4 (IgG4)-related disease [64]. Moreover, Protein Kinase Cε (PKCε) inhibition resulted in a decrease in collagen I protein and activated TGFβ levels in a rat model of heart fibrosis [65]. In human pulmonary fibroblasts, *OGR1* overexpression was demonstrated to reduce Smad2 phosphorylation in comparison to wild type fibroblasts (both treated with TGFβ) resulting in reduced *COL1A1* mRNA levels [66]. In contrast, *Ogr1* expression was found associated with increased *Col1a1* mRNA in a mouse model of fibrotic lesions in Crohn’s disease suggesting a context-dependent mechanism of OGR1 function [63,66]. Supporting this is the observation that in a dextran sodium sulphate-induced colitis mouse model, decreased *Col1a1* mRNA, decreased collagen protein deposition and decreased hydroxyproline content were noted following *Ogr1* knock-out in comparison to the *Ogr1* wild-type controls [63]. Moreover, the osteopontin-derived RGDSLAYGLR peptide, generated following cleavage by thrombin, was found to activate the TGFβ receptor, leading to increased Smad phosphorylation and *Col1a1* mRNA levels in a mouse model of pressure overloaded heart [67].

The TGFβ pathway induces a negative feedback loop downregulating its downstream activated genes including *COL1A1*. This feedback loop is mediated by NR4A1; a member of the steroid/thyroid hormone receptor subfamily [42,43]. This transcription factor represses transcription of TGFβ target genes, including *COL1A1* and *COL1A2*. For *COL1A1*, NR4A1 recruited Sp1, swi-independent-3 transcription regulator family member A (SIN3A), REST corepressor-1 (RCOR1 or CoREST, REST denoting RE1 Silencing Transcription Factor), histone deacetylase 1 (HDAC1) and Lysine(K)-specific demethylase 1A (KDM1A or LSD1) to an Sp1 binding site at -242 bp, repressing *COL1A1* transcription in dermal fibroblasts from systemic sclerosis patients. Knock-down of any of the aforementioned members of the repressor complex in the same system prevented NR4A1-mediated repression of TGFβ target genes [43]. This NR4A1 repressor function was also suggested in respective experiments using human endometrial stromal cells [42]. A repressor role for *COL1A1* transcription has also been attributed to KLF6 through binding to GC boxes of the gene promotor, decreasing the fibrogenic activity of mouse hepatic stellate cells (HSC) [48].

Beyond the TGFβ pathway, an influence of male and female sex hormones on collagen regulation has been suggested based on observed differences in the incidence of fibrosis between males and females [68]. In testosterone deficient mice, injected testosterone and oestrogen were able to reduce *Col1a1* mRNA expression leading to a less pronounced fibrosis phenotype [69]. In the context of endometriosis-related fibrosis, primary human peritoneal mesothelial cells in which Sp/KLF transcription factor Krüppel-like factor 11 (KLF11) was knocked down, treatment with progesterone resulted in an increase in *COL1A1* mRNA levels, whereas conversely a decrease of the *COL1A1* mRNA levels following treatment with oestrogen was noted. KLF11 knock-down by itself did not result in any significant changes in *COL1A1* mRNA expression [68]. If these observations are confirmed in additional studies, they may form the basis for sex-specific recommendations for disease screening and early diagnosis as well as provide a window of opportunity for hormone therapy for fibrotic diseases [68,69].

The c-Myb transcription factor was shown to indirectly regulate COLα1 (I), as a reduction of the levels of COLα1 (I) was observed in a *c-Myb* knock-out rat model of cardiac fibrosis (as compared to the wild type rat model of fibrosis). It remains unclear whether c-Myb targets the *Col1a1* genes directly, thus resulting in reduced mRNA production, or other factors are implicated [49].

Similarly, *COL1A1* mRNA levels were found to be increased in blood samples of systemic sclerosis patients and skin fibroblasts from these patients transfected with *NOTCH2.* NOTCH2 is a receptor, whose intracellular domain migrates to the nucleus and forms a transcriptional activator complex with Recombination signal binding protein for immunoglobulin kappa J region (RBPJ, also known as RBPSUH) upon receptor activation. Whether NOTCH2 targets *COL1A1* directly or indirectly remains to be elucidated [50].

Hypoxia inducible factor 1 subunit alpha inhibitor (HIF1AN), which hydroxylates the transcription factor Hypoxia Inducible Factor 1 Alpha (HIF1α), has been found to decrease *Col1a1* mRNA levels in fibroblast cell lines derived from rat kidney. HIF1AN is known to inhibit various important transcriptional regulators, including HIF1α, IκΒ and NOTCH. *HIF1α* knock-down in renal epithelial cells also reduced COLα1 (I) levels. It is therefore hypothesized that HIF1AN acts as a repressor of *COL1A1* thus protecting from kidney fibrosis, although the underlying mechanism remains to be elucidated [51]. Some mechanistic evidence was provided is in a separate study on human chondrocytes, where HIF1α reduced *COL1A1* transcription by recruiting Sp3 [70].

Similarly, indirect evidence for the regulation of *Col1a1* by the YAP/TAZ proteins has been provided: transgenic mouse models of liver fibrosis overexpressing YAP proteins displayed an increase in collagen deposition and increased *Ctgf* mRNA levels [52]. Although evidence for a direct interaction of YAP with the *COL1A1* promotor exists in the transcription factor database TcoFbase [71], YAP has also been suggested to activate the TGFβ-Smad2/3 pathway in kidney fibrosis [72].

*COL1A1* and *COL3A1* mRNA as well as collagen cross-linking enzymes were found to be upregulated by the transcription factor Zinc Finger E-Box Binding Homeobox 1 (ZEB1) in the context of myocardial infarction. Regulation of this transcription factor was linked to miR-590-3p. Further studies to investigate the underlying mechanism are needed [53], with some limited evidence provided by previous studies suggesting both direct ac tivation of *COL1A1* [73,74], but also via activation of the TGFβ pathway [75].

Further transcription factors suggested to regulate *COL1A1* include NF1, ZBTB7B (also known as c-Krox), TNF-α and STAT3 [76,77,78]. Moreover, an adenovirus E1A enhancer-like element, a viral core enhancer motif (VCE), a CCAAT binding factor motif, a pyrimidine rich region and a Glucocorticoid-Response Element (GRE) have been identified upstream of the *COL1A1* gene [76]. Experimental evidence for the relevance of the latter has been provided through studies in human dermal fibroblasts where activation of the glucocorticoid receptor using dexamethasone reduced *COL1A1* mRNA levels in a reversible manner [79].

#### 2.2.2. COL1A1 Regulation on the Post-Transcriptional Level

Regulation on the post-transcriptional level can occur via stabilization of the mRNA [80], which is likely to increase protein synthesis, or induction of mRNA degradation (for example via miRNAs [81,82,83,84,85,86,87,88,89,90,91,92,93,94] or long non-coding RNAs (lncRNAs) [95]), likely reducing protein synthesis.

A direct impact on the turnover of the *Col1a1* mRNA has been found for Terminal Nucleotidyltransferase 5A (TENT5A, encoded by *Tent5a* in mice), which is a poly(A) polymerase of mRNA. The induced polyadenylation was found to prolong *Col1a1* mRNA turn-over time in osteoblasts and osteocytes when comparing wild type to a *Tent5a* knock-out mouse model [80].

Decreased *Col1a1* mRNA levels as a consequence of miRNA regulation has been suggested following *Lin28B* (a suppressor of miRNA synthesis) knock-out in a mouse model of alcohol-induced liver fibrosis [81]. By now, microRNAs miR-98 [82], miR-126-5p [83], miR-218-5p [83], miR-328-3p [84], miR-338-3p [85] and miR-29b-3p [86] were found experimentally to directly target *COL1A1* [82,83,84,85,86]. Links to *COL1A1* mRNA were observed mainly in the context of fibrosis [82,85,86], but also lung adenocarcinoma [83] and breast cancer [84].

In brief, a binding site for miR-98 was predicted upstream of *COL1A1*, which was confirmed in human hypertrophic scar fibroblasts. In this system, an inverse correlation of *COL1A1* mRNA expression levels and miR-98 was found [82]. Similarly, the increase in *COL1A1* mRNA levels in the presence of TGFβ was attenuated by miR-338-3p in a human lung fibroblast cell line, a human embryonic kidney cell line as well as in vivo in a bleomycin induced mouse model of pulmonary fibrosis. Interestingly, knock-down of circHIPK3 in the mouse model led to reduced *Col1a1* mRNA levels, nevertheless, the underlying mechanism and interplay with miR-338-3p were not investigated [85].

Members of the miR-29 family, namely, miR-29a, miR-29b and miR-29b-3p have also been implicated in the downregulation of *Col1a1* mRNA following administration of respective mimics in various fibrotic models. These include two mouse models of liver fibrosis and human HSC cells (miR-29a [88]), cultured rat heart cells and a rat model of cardiac fibrosis (miR-29-3b [87]), as well as mouse cardiac fibroblasts and a mouse model of diabetic cardiomyopathy (miR-29b-3p [86]). Interestingly, in the latter case, the same circular RNA (circHIPK3), implicated in the abovementioned regulation of *COL1A1* mRNA by miR-338-3p could also affect the *Col1a1* mRNA regulation by miR-29b-3p [86].

In the context of cancer, miR-126-5p and miR-218-5p were predicted and confirmed to downregulate *COL1A1* mRNA levels through binding sites in its 3′ untranslated region. These microRNAs were in turn downregulated by the lncRNA linc00511 in the tested model (lung adenocarcinoma cell lines) [83]. Similarly, miR-328-3p was found to reduce *COL1A1* mRNA levels in primary breast cancer cells, through binding to the mRNA 3′ untranslated region. miR-328-3p was in turn downregulated by hsa-circRNA-002178, a circular RNA molecule overexpressed in breast cancer and associated with worse prognosis [84].

Indirect evidence for the involvement of additional miRNAs in the regulation of *Col1a1* mRNA in the context of fibrosis has also been accumulated; these include miR-21 and miR-141, as tested in fibroblasts from systemic sclerosis patients (miR-21 [90]) or myofibroblasts in the context of chronic liver disease (miR-141 [91]). In both cases, the underlying hypothesis is that the impact on *Col1a1* mRNA was mediated via affecting PTEN [90,91].

Similarly, miR-326 was shown to directly interact with *TGFβ* mRNA and its levels were inversely correlated with the mRNA levels of, amongst others, *COL1A1* in primary endometrial stromal cells from patients having intra-uterine adhesions [92]. Along these lines, miR-23a and miR-28a were found to decrease COLα1 (I) protein levels in a CCl_4_ induced mouse model of liver fibrosis [93], and miR-1954 to reduce *Col1a1* mRNA levels in an angiotensin II-induced mouse model of cardiac fibrosis [94]. In both cases an indirect effect was hypothesized through affecting TGFβ (miR-23a and miR-28a [93]) or Thrombosponin-1 (Thbs1) (miR-1954) [94].

Aside from miRNAs, lncRNAs can influence both the mRNA and the protein levels of *Col1a1.* Specifically, knock-down of lncRNA Protein Folding Activity of the Ribosome (PFAR) prevented the increase of *Col1a1* mRNA levels after administration of TGFβ in mouse lung fibroblasts, in comparison to mouse lung fibroblasts not challenged with TGFβ. A decrease of COLα1 (I) levels was also observed [95]. Interestingly, in this system, the previously mentioned YAP transcription factor was mentioned as a downstream effector of PFAR [52,95].

#### 2.2.3. COLα1 (I) Regulation by Post-Translational Modifications and Cleavage

The post-translational modifications most frequently observed on COLα1 (I), as abovementioned, include proline and lysine hydroxylation (Figure 1). Lysine hydroxylation, followed by LOX-aided oxidative deamination, is vital in the formation of cross-links (Figure 2). Mutations of the LOX gene have been identified in Bruck syndrome, a rare, genetic disease with a phenotype similar to OI [4]. Moreover, changes in the expression of *COL1A1* mRNA levels in parallel to alterations in the mechanical stretching properties of periodontal osteoblasts were associated to changes in LOX mRNA levels [96].

COLα1 (I) proline hydroxylation is a post-translational modification of high functional significance, as its defects, either due to mutation in collagen genes and/or in the involved factors (for example in the gene encoding for the prolyl 3-hydroxylation complex) have been linked to a number of musculoskeletal diseases (of the most known being various forms of OI) [97]. As an example, in a mouse model where Pro986, the only known target for proline 3-hydroxylation on COLα1 (I), was mutated to Alanine (Ala), a mild increase in lysine hydroxylation and an extensive increase in glycosylation of COLα1 (I), contrasted by a decrease in COLα1 (I) deposition were observed in primary osteoblasts and fibroblasts [14]. These osteoblasts additionally showed delayed COLα1 (I) incorporation in collagen I and decreased collagen I fibril diameter, as well as decreased cross-linking. Along these lines, significant alterations in the skeleton of the homozygous Pro986Ala mice were observed (smaller and narrower rib cages, delayed ossification in the cranium and digits, more frail bones in comparison to controls) [14]. A similar phenotype has been observed in humans having a mutation in one of the three genes encoding the prolyl 3-hydroxylation complex (*CaRtilage-Associated Protein (CRTAP)*, *Prolyl-3-Hydroxylase1 (P3H1) and peptidyl-prolyl cis-transisomerase cyclophilin B (PPIB)*). In humans, such mutations are additionally also associated with increased lethality and delayed growth, possibly due to the chaperone role of the prolyl 3-hydroxylation complex [14].

COLα1 (I) post-translational modifications can also occur in the context of disease, for example N-homocysteinylation. Sites for N-homocysteinylation were identified on COLα1 (I) (Lys160, Lys266, Lys583, Lys1085 and Lys1225) in a mouse model of cystathionine β-synthase deficiency. In these cases, reduced collagen I deposition in the ECM as well as reduced cross-linking, despite unchanged levels of lysyl oxidase and unchanged collagen turnover being observed. Presumably, N-homocysteinylation blocks the lysine residues from cross-linking [98].

Disrupted collagen protein folding linked to reduced Glucose-Regulated Protein (Grp78) and protein disulfide isomerase (PDI), and subsequently to increased polyubiquitination and decreased collagen I deposition in the ECM has also been reported in primary rat HSCs (from fibrotic liver) [99].

In addition, failure to cleave off the signal peptide after export to the ECM, associated with a Gly22Arg mutation (Gly denoting glycine, Arg denoting arginine) in *COL1A1*, results in severe OI in humans. This mutation occurs at the edge the signal peptide (Figure 1), drastically reducing the splicing probability. Fibroblasts from a patient having this mutation, as compared to fibroblasts derived from a healthy individual, revealed a delayed pro-COLα1 (I) secretion and increased collagen I levels inside the cell [100]. In a different system, in particular human pancreatic adenocarcinoma, increased deposition of the pro-COLα1 (I) C-terminal propeptide in the extracellular matrix attributed to the cancer-associated fibroblasts and a decrease in collagen I fibrils was observed [101]. These effects could also be reproduced when mutating the cleavage site for BMP1 (Asp1219Arg, see also Figure 3), leading to a decrease in collagen I deposition, as well as an increase of pro-collagen I fibrils (both propeptides still attached) [101].

Further proteases besides the MMPs [36,37] involved in the proteolytic cleavage of (pro-)COL1α1 (I) after transport to the ECM and their regulators have also been implicated in alterations on COLα1 (I). As an example, Secretory Leukocyte Protease inhibitor (*Slpi*), which regulates serine proteases involved in the activation of MMP-2 and MMP-9, protected against an increase in *Col1a1* mRNA levels observed in wild type mice having bleomycin-induced pulmonary fibrosis in comparison to *Slpni* knock-out mice [102]. How exactly SLPI exerts its effect, through MMP-2 and MMP-9, remains to be elucidated, although evidence for the involvement of the TGFβ pathway exists, namely an upregulation of phosphorylated Smad2 in *Slpi* knock-out mice [102]. Similarly, tryptase-β was associated with increased COLα1 (I) deposition in a mouse model of chronic prostatitis also presenting with bladder fibrosis. This was attributed to the protease-activated receptor 2 (PAR2) as its knock-out prohibited the COLα1 (I) increased deposition with improvement on bladder function [103]. Collectively, these data reflect the importance of COLα1 (I) proteolytic cleavage in COLα1 (I) function.

### 2.3. COLα1 (I)-Cell Receptor Interactions

COLα1 (I) interacts with several cell receptors inducing diverse downstream signalling pathways. Examples are members of the integrin family [104,105,106,107], Endo180 [also known as Urokinase Plasminogen Activator Receptor-Associated Protein (uPARAP)] [108], the tyrosine kinase Discoidin Domain Receptors (DDR) [109] and Leukocyte-Associated Immunoglobulin-like Receptor (LAIR) [110]. In these interactions, hydroxyproline residues are frequently involved (interactions with integrins and DDRs) (e.g., [106]), and/or the PGP motif (e.g., interactions with the C-X-C chemokine receptors (CXCR [111,112,113]). In general, three major regions (major ligand-binding regions) on COLα1 (I) encompassing the protein domains ranging from amino acid (AA) 80–200, 680–830 and 920–1464 are known to interact with various receptors [114].

The interaction of COLα1 (I) with integrin β subunits of integrin cell receptors was found to be of importance for the transition of reactive astrocytes to scar-forming astrocytes in a spinal cord injury mouse model. Cultured reactive astrocytes derived from this model transitioned to scar-forming astrocytes only in the presence of collagen I coating: this transition was prevented in the presence of an anti-integrinβ-antibody, targeting the interaction between integrin β subunit and extracellular collagen I. Nevertheless, the specific type of integrin receptor amongst those expressed on astrocytes (integrin α1β1, integrin α2β1, integrin α10β1 and integrin α11β1) or the activated intracellular downstream pathways involved in this process were not described [105]. Similarly, ovarian cancer cell lines cultured in the presence of COLα1 (I) displayed increased migration and invasion potential, which was blocked when using a COLα1 (I)-specific antibody. Akt was suggested as implicated in this process, as it displayed an increase in phosphorylation in the presence of COLα1 (I), which was absent when integrinβ was knocked down or inhibited [107].

Endo180 is an endocytic mannose receptor mediating the uptake and internalization of collagen I (mainly when in denatured and fragmented form) into endosomes and eventually lysomes for final degradation. As such, it is considered to have a significant impact in collagen I remodelling and, through that, matrix architecture. As an example, association of reduced Endo180 with aberrations in collagen fibre formation during photoaging in dermis has been hypothesized [115]. Similarly, loss of structural organization of rat fibroblasts upon knocking out Endo180 could be observed; in this system the interacting region of COLα1 (I) with Endo180 was found to be in the region between AA 1000 and 1453, encompassing a part of the triple helix and the C-terminal propeptide of pro-collagen I (Figure 1) [108].

The interaction of the extracellular COLα1 (I), in its native triple helical structure, with the Discoidin Domain Receptors (DDR, specifically DDR1 and DDR2) has also been demonstrated [109]; the latter are receptor tyrosine kinases with collagens being their only known ligand [116]. Through this interaction, DDRs have been implicated in a variety of functions including fibrillogenesis and ECM remodelling [117]. As an example, intact COLα1 (I) was shown to promote DDR1 degradation in a mouse pancreatic adenocarcinoma cell line, whereas a COLα1 (I) fragment (from the N-terminus until and including Gly775) produced following MMP-8 cleavage, was found to activate the DDR receptor. This was proven by transfecting this pancreatic adenocarcinoma cell line with cleavage-resistant *COL1A1* (e.g., having mutations in *COL1A1* 776Pro; 777Pro or 774Pro and 776Met). The COLα1 (I)-DDR-induced downstream pathway was hypothesized to involve the activation of NFκB-NRF2 and mitochondrial biogenesis [109]. The interplay of the two factors was also further suggested by the fact that the growth-promoting impact of the COLα1 (I) on pancreatic cancer cells could be inhibited following the addition of DDR1 inhibitors or in the presence of cleavage-resistant COLα1 (I). Along these lines, mutations reducing collagen cleavage are associated with increased survival in human pancreatic adenocarcinoma patients [109].

Collagens are also known to interact with the leukocyte-associated immunoglobulin-like receptor-1 (LAIR-1, also known as Cluster of Differentiation (CD) 305), an immune checkpoint inhibitor found on, amongst others, leukocytes [110,118]. LAIR-1 interacts with the typical glycine-proline-hydroxyproline motif characteristic for collagens, including collagen I [118]. Moreover, collagen I fragments of the same molecular weight generated extracellularly by MMP-1 (C-terminal of AA 775) and MMP-9 cleavage (cleavage site not known) were able to inhibit T-cell activation and interferon (IFN)-γ secretion in vitro, this could be counteracted by a soluble decoy receptor LAIR-2 of apparent high affinity for collagen [38]. Tumours often overexpress LAIR-1, thereby evading the immune system. Increased collagen deposition has been associated with immune checkpoint inhibitor therapy resistance (in particular Programmed cell Death-1 (PD-1)/Programmed cell Death Ligand-1 (PD-L1) therapy). Given the function of LAIR-1 as an immune checkpoint inhibitor and its interaction with collagens, inhibiting both the increase of LAIR-1 and the increase in collagen I, in combination with PD-1/PD-L1 therapy, could have a beneficial effect in tumours expressing high levels of collagen in the tumour microenvironment. This was confirmed by reduced tumour growth and increased immune activation in a mouse model of breast and colon cancer, in case LAIR-1, TGFβR2 and PD-1/PD-L1 inhibitors were administered simultaneously, as compared to separately [110].

COLα1 (I) has been found at increased levels in many cancers with observed impact on various signalling pathways [21,119,120,121,122,123]. These include the Wnt-Planar Cell Polarity (PCP) pathway, an observation that was made in colorectal cancer cell lines where *COL1A1* knock-down resulted in suppression of factors from this pathway (including the Ras-related C3 botulinum toxin substrate 1—Guanosine TriPhosphate (Rac1-GTP), phosphorylated JNK and Ras homolog family member A-GTP (RhoA-GTP). As Frizzled and Receptor tyrosine kinase-like orphan receptor 2 (ROR2)/related to receptor tyrosine kinase (RYK) are known receptors of this pathway, they were suggested to interact with COLα1 (I), although this was not experimentally validated [21]. In breast cancer cell lines, a positive correlation between *COL1A1* and *CXCR4* has been observed; specifically, knocking down *COL1A1* reduced *CXCR4* mRNA expression as compared to wild type cells [121]. Members of the same family of receptors, namely CXCR1 and CXCR2, bind N-acetylated Pro-Gly-Pro motifs. These interactions were found to be involved in wound healing, neovascularization and immune cell infiltration in human endothelial progenitor cells and Peripheral Blood Mononuclear Cells (PBMCs), and cell migration as was found in cartilage endplate stem cells [113]. Collectively, some knowledge on COLα1 (I) interactors on the cell surface has been apparently accumulated. Nevertheless, information is still missing. For example, in breast cancer cells, COLα1 (I) has been found to induce the intracellular PI3K-Akt-mTOR pathway including also an activation of the transcription factor YAP, nevertheless, the specific COLα1 (I) extracellular receptor inducing these pathways remains to be elucidated [120]. Via affecting its receptors and pathways, COLα1 (I) has been associated with pleiotropic phenotypic impact including metastasis (in glioma, colorectal, gastric, breast and thyroid cancers [21,119,120,121,122,123]), immune infiltration (glioma [119]), cancer stem cell activation (e.g., breast cancer [120]) and others.

## 3. Conclusions: Outlook and Therapeutic Implications

COLα1 (I) synthesis and further processing is a complex process, including intracellular addition of various post-translational modifications such as hydroxylation of prolines and lysines (Figure 1) [1,14,15,16], leading to cross-linking (Figure 2) [1,3,15,26,27,28,29,30] and proteolytic cleavage (Figure 3) in the ECM [3,36,37], and ultimately in the deposition of mature fibrils [14]. This process is regulated at the epigenetic, transcriptional, post-transcriptional and post-translational level, all of which have been explored in the context of this review. It is acknowledged that relevant manuscripts may have been unintentionally omitted linked to the specific keywords used in our search; nevertheless, it may be safe to say that a representative overview of the current status is provided.

Compared to the other levels of regulation, limited evidence exists for the epigenetic regulation of *COL1A1* in disease, and the evidence compiled here was all in the context of fibrosis. Evidence was indirect for DNMTs [39,40,41], and direct for the involvement of HDACs and KDM1A induced by NR4A1 in the context of the TGFβ pathway [42,43] as well as recruitment of HDACs by KLF11 [68]. Moreover, indications of epigenetic impact of sex-specific differences on *COL1A1* have been identified, underscoring the need for further *COL1A1* study in a sex-specific manner [42,43,68]. More extensive evidence exists on the transcriptional level, in which the TGFβ pathway is highly implicated mainly through the regulation of Smad transcription factors binding directly to *COL1A1.* In addition, the role of NF-κB, AP2 and NR4A1 in *COL1A1* transcription is well established. Evidence for the involvement of other transcription factors (such as NOTCH2, c-Myb, HIF1α, β-catenin, ZEB1 and YAP) is indirect, and it remains to be confirmed if these transcription factors target *COL1A1* directly [49,50,51,52,68,75]. On the post-transcriptional regulation level, direct evidence for *COL1A1* mRNA regulation exists for TENT5A and six microRNAs (miR-98, miR-126-5p, miR-218-5p, miR-328-3p, miR-338-3p and miR-29b-3p) [80,82,83,84,85,86]. However, for at least the same number of post-transcriptional regulators identified (miR29a, miR-29, miR-21, miR-141, miR-1954, PFAR), only indirect evidence for their involvement in *COL1A1* mRNA regulation exists [81,87,88,89,90,91,94,95]. The latter in many cases involves regulation of other proteins, for example PTEN by miR-21 and miR-141 [90,91], while the way such regulators implicate *COL1A1* mRNA remains to be elucidated. Most evidence on these regulatory mechanisms was provided in the context of fibrosis-related diseases. On the post-translational level, most evidence is direct [14,95,96,98,99,100,101], involving altering the post-translational modification of collagen I (proline hydroxylation [14], N-homocysteinylation [98] or mutations altering proteolytic cleavage sites of pro-collagen I [100,101]). In addition, involvement of proteases such as MMP-2 and tryptase β has been implicated in COLα1 (I) regulation in the context of various diseases [102,103].

COLα1 (I) itself interacts with various cell receptors (integrinβ, Endo180, DDR, LAIR-1, CXCR4 and CXCR2) with, in general, limited knowledge on the induced downstream pathways [104,105,106,107,108,109,110,111,112,113,114,121,123]. Here, the need for further experimental evidence on the involved regions on COLα1 (I) and the induced signalling pathways downstream of the receptors is underscored.

Of note, the presented regulators and interactors were studied in different disease contexts and moreover, even within the same disease occasionally in different models (animals, primary cell culture, cell lines), for which the disease can be induced in different ways, resulting in most cases in the induction of different molecular pathways (for example CCl_4_ [39,40,41] and alcohol induced liver fibrosis [39], revealing different involved factors in each case e.g., DNMT1 and DNMT3A in the former [39,40,41], Lin28B in the latter [81]).

Aside from the differences in regulation, the role of collagen I in diseases is also pleiotropic. The majority of the diseases featured in this review can be grouped into three major categories. The first category consists of fibrosis and fibrosis-related diseases. While various factors (such as the aforementioned DNMT1, DNTM3A, Lin28B and others) have been linked to fibrosis, in most cases, it remains unclear how they are activated [18,39,40,41,42,46,47,49,50,51,52,53,55,59,60,63,65,67,68,81,82,84,85,86,87,88,90,91,92,93,94,95,99,102,104]. Another major category is the bone-related diseases, mainly represented by OI. Here, mutations are affecting the stability of the collagen I molecule, which eventually leads to reduced bone stability [4,14,80,100]. The last major group of diseases consists of various cancer forms, for which the COLα1 (I) chain has been associated with metastasis, immune infiltration and cancer stem cell activation. For the various types of cancer featured in this review, collagen I interacts with various receptors on cancer cells (including integrinβ, Endo180 or uPARAP, DDR, LAIR-1, CXCR2 and CXCR4), opening interesting perspectives for the development of new drugs targeting such interactions [21,41,83,108,109,110,120,121,123].

In conclusion, currently, the pathways regulating *COL1A1* gene and mRNA expression as well as (pro-)COLα1 (I) production and maturation are relatively well elucidated in some cases such as TGFβ, rendering them better candidates for further consideration in drug development. In fact, clinical trials focusing on COLα1 (I) have been organized (for example in the context of fibrosis, [124], or OI, [125], NCT02531087) and several more are ongoing (see for example [126]). Nevertheless, a need for further investigation of COLα1 (I) associated molecular pathways is clearly needed. In this effort, investigation of a particular pathway across different models is required to define commonalities, as was performed on a small scale in the case of linc00511 [83] and miR-29a [88]. The need for further characterizing the interaction of COLα1 (I) with cell receptors is also underscored with obvious potential therapeutic impact, especially in the context of cancer, given the association of such interactions with metastasis [21,119,120,121,122,123]. In all cases, investigation of the involved molecular process in a more spherical manner linking the currently disparate different levels of information is required to better establish the therapeutic implications of *COL1A1* and COLα1 (I) regulation.

## Figures and Tables

**Figure 1 ijms-24-10004-f001:**
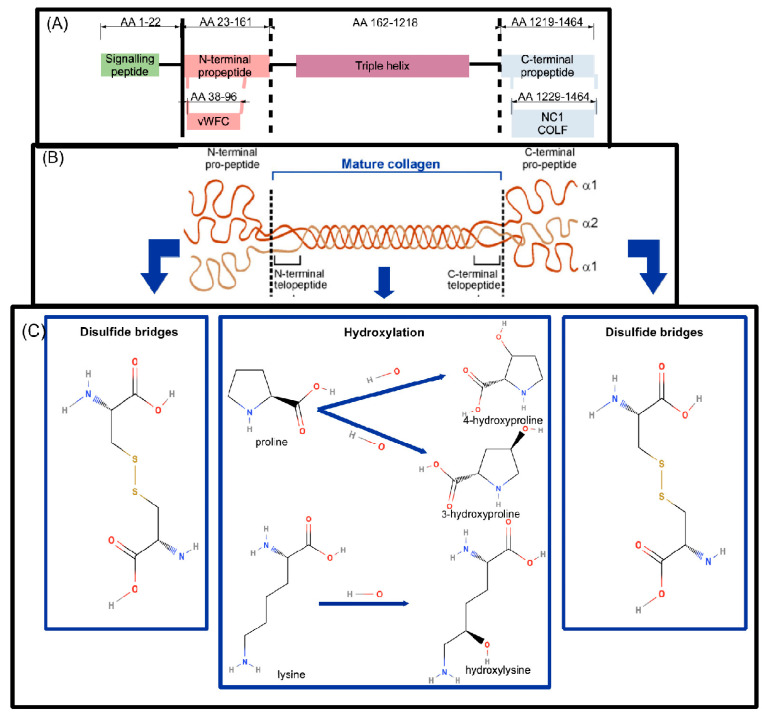
Collagen type I alpha 1 (COLα1 (I)) domains and post-translational modifications implicated in collagen maturation. (**A**) The different parts of the COLα1 (I) primary structure are shown: a signalling peptide (green), an N-terminal propeptide (pink), triple helix (purple) and C-terminal propeptide (light blue). On the second row, specific domains of the propeptides are indicated (vWFC: von Willebrand Factor type C domain, NC1: Fibrillar collagen C-terminal non-collagenous domain). Mutations at various positions throughout the protein sequence have been associated with bone diseases (evidence from UniProt), whereas regions aa 80–200, 680–830, 920–1464 have been involved in interactions with various receptors (Section 2.3). (**B**) Pro-collagen I with the N- and C-terminal peptides, flanking the COLα1 (I) chain, are shown. (**C**) The most common modifications of N-terminal pro-collagen I (left) and C-terminal pro-collagen I (right) as well as the most frequent modifications of the COLα1 (I) triple helix (centre) are shown. The upper part of the figure was generated in-house. The middle part of the figure, showing the COLα1 (I) chain and propeptides, was derived from Fan et al. [6] (CC-BY2.0 Licence). Bottom line figures were generated with structures from PubChem database ([7], specifically [8,9,10,11,12,13]), based on the post-translational modifications outlined above [1,14,15,16]. Abbreviations are explained in Appendix A, Table A1.

**Figure 2 ijms-24-10004-f002:**
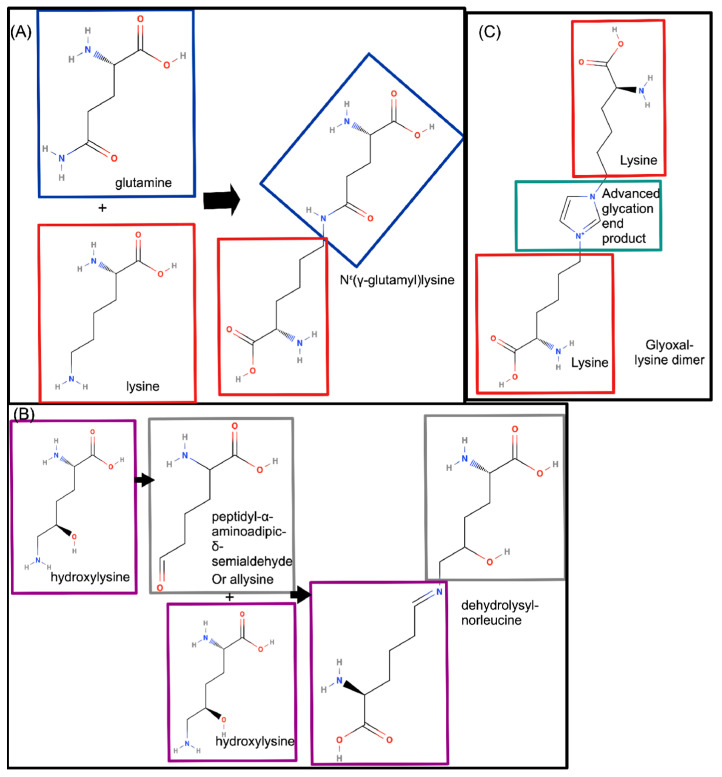
Types of cross-linking observed in type I collagen (collagen I). (**A**) Cross-linking via an N^ε^(γ-glutamyl)lysine residue, which links a glutamine residue (blue rectangle) and a lysine residue (red rectangle) [26]. (**B**) Cross-linking of hydroxylysine residues (purple rectangles), by oxidative deamination of one hydroxylysine to peptidyl-α-aminoadipic-δ-semialdehyde (grey rectangle), followed by a spontaneous condensation reaction with the other hydroxylysine [28]. (**C**) Cross-linking through advanced glycation end products (green rectangle), linking two lysine residues (red rectangles) [27]. On each of the final chemical structures, the colour-coded rectangles denote the part of the molecule derived from the respective parent molecules. Chemical structures were retrieved from PubChem database ([7], specifically [9,12,31,32,33,34,35]).

**Figure 4 ijms-24-10004-f004:**
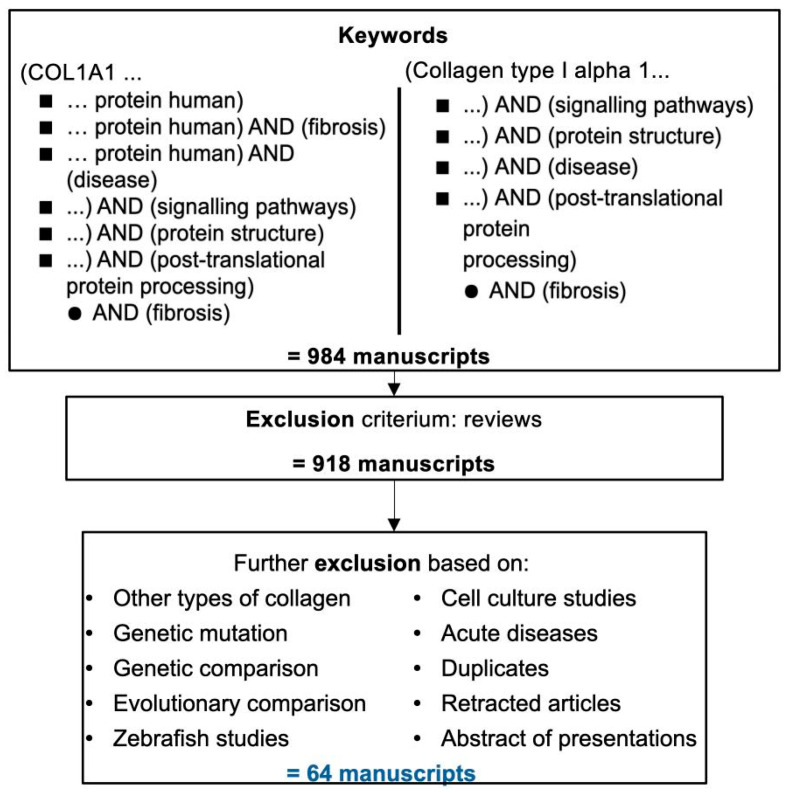
Summary of the workflow for this review. MeSH term “COL1A1” was combined with every search term in the first column, and “Collagen type I alpha 1” was combined with all terms in the second column. In addition, a search using the term ‘COLα1 (I)’ was also performed and furthermore, similar articles for “Collagen cross-links in mineralizing tissues: a review of their chemistry, function, and clinical relevance” listed in PubMed were retrieved. The complete oversight of all search results retained per search query, can be found in Appendix A, Table A2. Appendix A, Table A3 also lists the manuscripts forming the backbone of this review.

**Figure 5 ijms-24-10004-f005:**
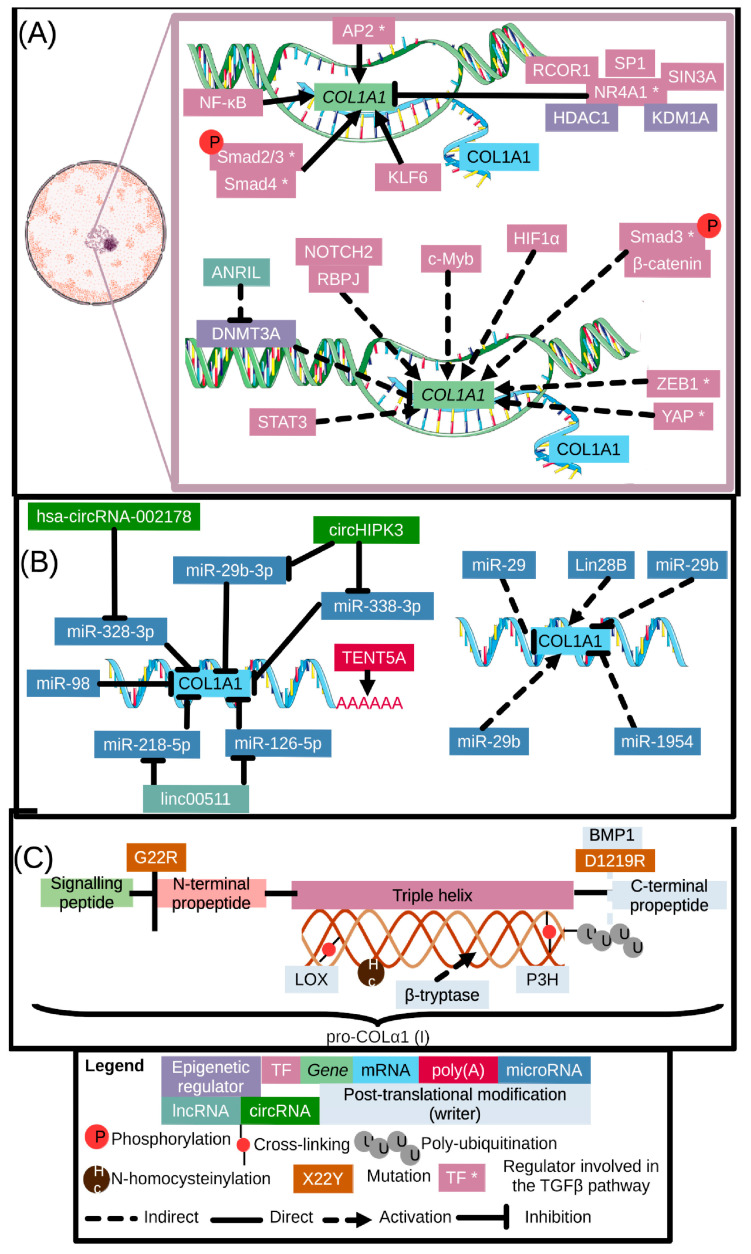
Summary of the different regulators of COLα1 (I) on the epigenetic as well as transcriptional (panel (**A**)), post-transcriptional (panel (**B**)) and post-translational level (panel (**C**)). Regulators are color-coded in accordance with their levels of regulation, and an asterisk (regulator *) following a regulator denotes links of the specific factor to the TGFβ-pathway. Full lines signify direct interactions, dashed lines indirect links. Arrows denote upregulation, while T-shaped arrows denote inhibition (at least in the systems described in the main text). A more elaborate summary of our findings can be found in Appendix A, Table A4, which lists all regulators, type of regulation and whether they are direct or indirectly linked to COLα1 (I), as well as the respective references and the diseases and models used to investigate this regulation. Image made partly using Servier Medical Art, provided by Servier, licensed under a Creative Commons Attribution 3.0 unported license (nucleus, DNA, transcription, RNA). Regulation by proteases is shown in more detail in Figure 3 and a list of abbreviations can be found in Appendix A, Table A1.

## Data Availability

Data sharing not applicable.

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
