# Peer review of "Reviewing the Regulators of COL1A1"

_ijms, 2023, doi:10.3390/ijms241210004_

Round 1

Reviewer 1 Report

The paper provides a comprehensive review of regulators of COL1A1 in disease, and overall, it is well-written. However, I suggest the following changes to improve its readability:   

1.In the "2.1 Literature Search" section, using the keyword "col1a1 protein human" may cause relevant studies on col1a1 in animal models to be missed.

2. In section 2.1.3, where proteolytic cleavage in L579-599 and L606-613 is discussed, it may be more readable to combine these into one paragraph.

3.  Fig A1, which illustrates COL1A1 signal pathways, would be more useful if it were moved from the appendix to the main manuscript and placed in section 2 on molecular mechanisms.

4. Including a summary table or figure of the mutation sites of COL1A1 in section 2.3 on COL1A1-regulated pathways in disease would make it more accessible to readers.

5. This review article discusses the regulators of COL1A1 in disease. Table A4 provides a detailed summary of the main regulators of COL1A1. Including Table A4 in the main manuscript would enhance its accessibility and usefulness to readers.

6. It would be helpful to add a figure or table that summarizes the mutation sites of COL1A1 in section 2.3 to further illustrate the COL1A1regulators in disease."

7.Please double check the orders of sections, such as L206 2.2 Regulators of COL1A1, L225 2.1.1. COL1A1 gene epigenetic and transcriptional regulators, L438 2.1.2. COL1A1 regulation on the post-transcriptional level, L532 2.1.3. COL1A1 regulation by post-translational modifications and cleavage and L725 4. Conclusion: outlook and therapeutic implications.

Author Response

Reviewer 1:

The paper provides a comprehensive review of regulators of COL1A1 in disease, and overall, it is well-written. However, I suggest the following changes to improve its readability:    

Point 1. In the "2.1 Literature Search" section, using the keyword "col1a1 protein human" may cause relevant studies on col1a1 in animal models to be missed.

Reply 1: We would like to thank the reviewer for their thoughtful comments. However, we feel that the use of the keyword ‘human’ is warranted to filter out results without any clear link to human pathology, as human pathology is the focus of this review.

Point 2. In section 2.1.3, where proteolytic cleavage in L579-599 and L606-613 is discussed, it may be more readable to combine these into one paragraph.

Reply 2: We thank the reviewer for their attentive reading and have changed this accordingly.

Point 3.  Fig A1, which illustrates COL1A1 signal pathways, would be more useful if it were moved from the appendix to the main manuscript and placed in section 2 on molecular mechanisms.

Reply 3: We have moved this particular figure to section 2, near the section describing TGFβ-mediated regulation of COL1A1.

Point 4. Including a summary table or figure of the mutation sites of COL1A1 in section 2.3 on COL1A1-regulated pathways in disease would make it more accessible to readers.

Reply 4: We thank the reviewer for this comment. We investigated in more depth mutation sites of collagen 1 a1 (symbolize the protein as it should be) associated with diseases ;  as these are many throughout the protein sequence, relating mainly to bone diseases, to avoid confusing the reader, we opted to add a respective statement in Figure 1 legend. In addition, in the same legend we marked the regions of the protein associating with receptors, as described in section 2.3.

Point 5. This review article discusses the regulators of COL1A1 in disease. Table A4 provides a detailed summary of the main regulators of COL1A1. Including Table A4 in the main manuscript would enhance its accessibility and usefulness to readers.

Reply 5: We recognize the clarifying value of table A4. However, as table A4 is quite extensive and detailed, we consider it better to keep this table in the addendum, as we fear tables of this length might confuse the reader.

Point 6. It would be helpful to add a figure or table that summarizes the mutation sites of COL1A1 in section 2.3 to further illustrate the COL1A1regulators in disease."

Reply 6: We thank the reviewer for this suggestion, as mentioned above (point 4) a table listing the mutation sites has been added

Point 7.Please double check the orders of sections, such as L206 2.2 Regulators of COL1A1, L225 2.1.1. COL1A1 gene epigenetic and transcriptional regulators, L438 2.1.2. COL1A1 regulation on the post-transcriptional level, L532 2.1.3. COL1A1 regulation by post-translational modifications and cleavage and L725 4. Conclusion: outlook and therapeutic implications.

Reply 7: We thank this reviewer for their attentive reading, and have changed the numbering accordingly.

Reviewer 2 Report

The authors of a manuscript entitled: “Reviewing the regulators of COL1A1 in disease” present their summary of complex processes associated with regulating the expression of collagenous proteins, most notably collagen I. The paper, however, includes many errors, including in nomenclature and understanding of collagen-associated processes. Because of a relatively large number of these errors, only some are described:

1. The authors use inconsistent, confusing, and sometimes incorrect nomenclature to describe collagen protein and various gens. This reviewer suggests the following names:

a. To describe type I collagen, use “collagen I”

b. To describe collagen I alpha 1 chain, use alpha (a Greek letter)1(I).

c. To describe collagen I alpha 2 chain, use alpha (a Greek letter)2(I)

d. Corresponding procollagen I chains should have the “pro” prefix.

e. Use proper codes for the collagen I genes. for the human versions, COL1A1 and COL1A2; for the mouse versions Col1a1 and Col1a2.

f. Check symbols of other mentioned genes; e.g. the symbol for the gene encoding alpha SMA is ACTA2, not aSMA.

            Still, the authors may propose custom abbreviations of the above and other molecules, paying attention not to confuse proteins with genes or mRNA. For example, they should not use “COL1A1” to describe collagen I protein.

2. The authors should also check the text carefully to reflect the proper definition of collagen assemblies. for instance, lane 48-51: “The type of chains interacting to form a fibril depends on the collagen type, with the most abundant COL1 fibrils consisting of two collagen type I alpha 1 (COL1A1) and one collagen type I alpha 2 (COL1A2) chains (Figure 1), whereas the other fibrillar collagens are homotrimers.” The authors should distinguish between triple-helical collagen molecules (consisting of collagen alpha 3 chains) and collagen fibrils (consisting of many triple-helical collagen molecules).

3. The authors should update their knowledge on many collagen I-associated points. For instance, lane 106-110, they state, “lately COL1A1 homotrimers have also been detected. Such homotrimers are associated with structural and functional differences to the classical heterotrimer forms, for example an increased resistance against proteases [16], weaker intermolecular interactions leading to reduced tensile strength [17], and an increased lateral space [18].” Indeed, the existence of collagen I homotrimer is not a recent discovery but classic knowledge.

In another example, lanes 158-161, the authors claim that N and C proteinases cleave collagen I. These enzymes cleave procollagen variants of many fibrillar collagens.

4. Considering the disease aspect of this review, the authors should either focus on a specific group of collagen-related diseases, e.g., fibrotic disorders or heritable disorders, or indicate differences among the disease groups they selected.

5. The authors must indicate whether the processes they describe occur inside or outside cells.

Author Response

Reviewer 2:

The authors of a manuscript entitled: “Reviewing the regulators of COL1A1 in disease” present their summary of complex processes associated with regulating the expression of collagenous proteins, most notably collagen I. The paper, however, includes many errors, including in nomenclature and understanding of collagen-associated processes. Because of a relatively large number of these errors, only some are described:

Point 1. The authors use inconsistent, confusing, and sometimes incorrect nomenclature to describe collagen protein and various gens. This reviewer suggests the following names: 

  1. To describe type I collagen, use “collagen I”
  2. To describe collagen I alpha 1 chain, use alpha (a Greek letter)1(I).
  3. To describe collagen I alpha 2 chain, use alpha (a Greek letter)2(I)
  4. Corresponding procollagen I chains should have the “pro” prefix.
  5. Use proper codes for the collagen I genes. for the human versions, COL1A1 and COL1A2; for the mouse versions Col1a1 and Col1a2.
  6. Check symbols of other mentioned genes; e.g. the symbol for the gene encoding alpha SMA is ACTA2, not aSMA.

            Still, the authors may propose custom abbreviations of the above and other molecules, paying attention not to confuse proteins with genes or mRNA. For example, they should not use “COL1A1” to describe collagen I protein.

Response 1: We would like to thank the reviewer for their thorough explanation of the terminology and nomenclature of collagens, and recognize the importance of using the correct terminology. Thus, this have been changed accordingly in the review (throughout all sections).

Point 2. The authors should also check the text carefully to reflect the proper definition of collagen assemblies. for instance, lane 48-51: “The type of chains interacting to form a fibril depends on the collagen type, with the most abundant COL1 fibrils consisting of two collagen type I alpha 1 (COL1A1) and one collagen type I alpha 2 (COL1A2) chains (Figure 1), whereas the other fibrillar collagens are homotrimers.” The authors should distinguish between triple-helical collagen molecules (consisting of collagen alpha 3 chains) and collagen fibrils (consisting of many triple-helical collagen molecules). 

Response 2: We thank the reviewer for clarifying this point, and have changed this accordingly throughout all sections of the review.

Point 3. The authors should update their knowledge on many collagen I-associated points. For instance, lane 106-110, they state, “lately COL1A1 homotrimers have also been detected. Such homotrimers are associated with structural and functional differences to the classical heterotrimer forms, for example an increased resistance against proteases [16], weaker intermolecular interactions leading to reduced tensile strength [17], and an increased lateral space [18].” Indeed, the existence of collagen I homotrimer is not a recent discovery but classic knowledge.

In another example, lanes 158-161, the authors claim that N and C proteinases cleave collagen I. These enzymes cleave procollagen variants of many fibrillar collagens. 

Response 3: We have changed these sentences, to reflect this. We specifically used terminology as described in Van Huizen et al. [1].

Point 4. Considering the disease aspect of this review, the authors should either focus on a specific group of collagen-related diseases, e.g., fibrotic disorders or heritable disorders, or indicate differences among the disease groups they selected.

Response 4: This is a well taken suggestion however based on the keywords used in the literature search a broad overview of the role of collagen I was targeted. To address the comment, a paragraph was added to the conclusion of this review, comparing the information on the different diseases as retrieved in this review.

Point 5. The authors must indicate whether the processes they describe occur inside or outside cells.

Response 5: We have added clarifications for this aspect throughout the review.

  1. van Huizen, N.A.; Ijzermans, J.N.M.; Burgers, P.C.; Luider, T.M. Collagen Analysis with Mass Spectrometry. Mass Spectrom. Rev. 2020, 39, 309–335, doi:10.1002/mas.21600.

Round 2

Reviewer 1 Report

Dear authors, 

Thank you for addressing most issues raised. However, the present study has some major concerns that should be addressed before considering it for publication. 

1)In the manuscript, in the "2.1 Literature search" section, COL1A1 has been replaced by COLα1 (I ). There are many studies using word “COL1A1”. Would this affect the search results? And in Figure 1, “COL1A1” was still used. 

2)Please confirm whether Line 43 belongs to paragraph 1 or paragraph 2. 

3) Section 2.3 of the manuscript focuses on COLα1's receptor, however, it did not discuss the signal transduction pathway. Could you revise this section to provide a more balanced discussion of receptor and signal transduction? 

4)Please double-check Line 243 for accuracy regarding "Regulators of COL1A1 and COLα1 (I)." 

5)It would be helpful to reorganize the "2.1.1 COL1A1 gene epigenetic and transcriptional regulators" section with subheadings for different regulators such as TGFβ, NF-κB, NOTCH2, etc. 

6)Please double-check the reference style for consistency. 

Minor editing of English language required

Reviewer 2 Report

Although the authors have improved the nomenclature of collagen-coding genes and elements of collagen molecules, the manuscript still requires significant changes. In essence, successfully implementing these changes requires authors to revise their manuscript to ensure they fully understand the processes they describe. As presented in the current form, the review is a collage of collagen I-related information that lacks clear organization and definition of the leading theme.

Suggest editing by an English-proficient editor.

Round 3

Reviewer 1 Report

Thank you to the authors for addressing the raised issues. I think this manuscript is suitable for publication.